# Microglia in Brain Aging and Age-Related Diseases: Friends or Foes?

**DOI:** 10.3390/ijms262311494

**Published:** 2025-11-27

**Authors:** Kentaro Ishikawa, Risako Fujikawa, Kayoko Okita, Fumika Kimura, Takuya Watanabe, Shutaro Katsurabayashi, Katsunori Iwasaki

**Affiliations:** Department of Neuropharmacology, Faculty of Pharmaceutical Sciences, Fukuoka University, 8-19-1 Nanakuma, Jonan-ku, Fukuoka 814-0180, Japantwatanabe@fukuoka-u.ac.jp (T.W.); shutarok@fukuoka-u.ac.jp (S.K.); iwasakik@fukuoka-u.ac.jp (K.I.)

**Keywords:** microglia, aging, frailty, Alzheimer’s disease, cognitive function, inflammation, phagocytosis, lipid, CD11c, TREM2

## Abstract

With the global rise in population aging, establishing effective strategies for the prevention and treatment of age-related neurodegenerative diseases, as well as their prodromal stage of cognitive frailty, has become an urgent challenge. Recent studies have revealed that the neural basis of both frailty and age-related disorders is closely associated with chronic neuroinflammation and impaired clearance of cellular debris, processes that are primarily regulated by microglia, the resident immune cells of the brain. As aging progresses, microglia exhibit reduced surveillance and motility, diminished phagocytic efficiency, and transition into a proinflammatory, hyperresponsive state. Such maladaptive microglia contribute to synaptic loss, white matter deterioration, and the spread of neurodegenerative pathology. Conversely, single-cell transcriptomic studies have identified distinct microglial subsets, including CD11c^+^ microglia, which show upregulation of lysosomal and lipid metabolism pathways, enhanced debris clearance, and elevated neurotrophic factor expression. These features suggest that certain microglial populations adopt protective or adaptive phenotypes that preserve neural integrity. However, under chronic inflammation or pathological conditions, even protective microglia may become inflammation-promoting. This review summarizes current evidence on microglial changes in aging, frailty, and neurodegeneration, emphasizing their dual roles and discussing strategies that modulate microglial function to maintain brain health and prevent or treat frailty and age-related diseases.

## 1. Introduction

When considering microglia, should we view them as friends or as foes? Microglia, the resident immune cells of the central nervous system (CNS), play indispensable roles in maintaining brain homeostasis through immune surveillance and the clearance of cellular debris [1,2]. Under physiological conditions, microglia exhibit a highly dynamic and ramified morphology, constantly monitoring the neural environment to preserve tissue integrity [3]. However, with advancing age, microglia undergo profound morphological and functional alterations, including chronic activation, reduced motility, and dysregulated phagocytic activity [4,5,6]. These age-related changes lead to the establishment of a proinflammatory milieu and impaired clearance of waste products, thereby exerting deleterious effects on the nervous system and acting as “foes.” Indeed, abnormal activation and functional decline of microglia have been implicated in the pathogenesis of age-related neurodegenerative diseases, including Alzheimer’s disease (AD), the most common form of dementia [7,8]. Thus, understanding how microglia change with age is crucial for developing effective preventive and therapeutic strategies against such disorders. These age-dependent alterations in microglial function have also been recognized as key contributors to cognitive frailty, a condition defined by the coexistence of physical frailty and mild cognitive impairment (MCI). Cognitive frailty represents a potentially reversible state of vulnerability that precedes irreversible neurodegeneration [9,10]. While frailty was initially defined by physical decline, the concept has recently been expanded to include cognitive impairment, positioning it as a bridge between normal aging and neurodegenerative diseases [11]. Elucidating how aging microglia contribute to this transition may therefore be essential for preventing the progression from cognitive frailty to age-related neurodegenerative conditions.

Meanwhile, recent advances in single-cell transcriptomic analyses have revealed microglial subpopulations that behave as “friends” within the aged or diseased brain [12]. Among these are CD11c^+^ microglia and other heterogeneous subsets that cannot be explained by the traditional M1/M2 polarization framework. These subtypes contribute to debris clearance and tissue repair, thereby conferring resistance against age-related pathologies, although under certain conditions they can also elicit inflammatory responses and exacerbate neuronal injury. Thus, microglia possess a dual nature, acting as both allies and adversaries in the aging brain, and regulating or reprogramming their “friend-like” functions may hold the key to combating age-related neurodegenerative diseases.

In this review, we focus on the alterations of microglia that occur during aging, outlining both their pathological and protective aspects. We first provide an overview of how aging microglia may act as foes that promote neuroinflammation and neurodegeneration. We then highlight the friend-like properties of microglia, emphasizing protective subsets such as CD11c^+^ microglia that may contribute to tissue repair and neural resilience [13,14]. Subsequently, we summarize key molecular pathways that regulate beneficial microglial states and, finally, discuss translational perspectives, including insights from recent TREM2-targeted clinical trials and future directions for harnessing microglia to maintain brain health. Taken together, this review aims to provide an integrative perspective on whether microglia act as friends or foes in the aging brain and to explore emerging preventive and therapeutic strategies targeting microglia to counteract cognitive frailty and the progression of neurodegenerative diseases.

## 2. Aging Microglia as Foes

Microglia are known to undergo diverse functional and morphological alterations during aging. Understanding the roles of microglia in the aging brain may provide novel insights into preventing the transition from cognitive frailty to neurodegenerative diseases. In this section, we focus on the “foe-like” transformations of microglia that contribute to the progression of cognitive frailty and disease, and summarize the current knowledge in this field.

First, we outline age-related functional changes in microglia from four perspectives: (i) environmental surveillance within the central nervous system (CNS), (ii) migration toward sites of injury, (iii) cytokine production and inflammatory responses, and (iv) phagocytic activity toward cellular debris and Aβ (Figure 1). Furthermore, we highlight findings from frailty model mice, discussing how microglial dysfunction may determine whether brain aging remains reversible or progresses to irreversible neurodegeneration.

### 2.1. Surveillance Function (Environmental Monitoring and Process Motility)

Under physiological conditions, microglia constantly extend and retract their highly ramified processes to patrol the surrounding brain environment and maintain homeostasis [3]. In vivo imaging demonstrates that microglia in the young mouse brain exhibit rapid and dynamic process motility, whereas those in aged mice show reduced branching complexity and markedly slower process dynamics [15]. These alterations suggest that aging diminishes microglial surveillance capacity and restricts their ability to detect subtle changes in the microenvironment. Similar age-related morphological deterioration has been documented in the human brain, where microglia progressively acquire dystrophic features, including shortened, fragmented processes and reduced branching density [16]. This conserved structural decline likely delays the detection of fine neuronal or environmental signals, thereby compromising central nervous system homeostasis.

### 2.2. Migratory Function (Directional Movement Toward Injury Sites)

Microglia normally migrate rapidly toward sites of tissue damage to initiate inflammatory and reparative responses. In young adult mice, microglia extend their processes and accumulate at laser-induced lesions within minutes in a coordinated manner [3,17]. In contrast, aging markedly delays this response: microglia in aged mice show slower initiation of movement, reduced migration speed, and incomplete accumulation at injury sites [15,18]. In vivo retinal imaging similarly demonstrates delayed and attenuated migration in older animals [15]. These findings indicate that aging diminishes both the chemotactic velocity and sensitivity of microglia, resulting in slower and less effective responses to tissue injury.

### 2.3. Inflammatory Response

Aging drives microglia toward a primed state characterized by elevated basal expression of pro-inflammatory mediators and exaggerated responses to secondary stimuli [19,20,21]. Microglia from aged mice release higher levels of TNF-α and IL-6 even at rest and produce markedly greater amounts of IL-1β upon LPS stimulation compared with young microglia [20,22,23,24]. This heightened reactivity contributes to chronic low-grade neuroinflammation and accelerates tissue aging through senescence-associated mechanisms. Microglia in the aged brain also display hallmark features of cellular senescence, including DNA damage, telomere shortening, and lipofuscin accumulation, together with enhanced secretion of SASP-like cytokines such as IL-6 and MMPs [25,26,27,28,29].

In parallel, the anti-inflammatory capacity of microglia declines with age. For instance, microglia in aged brains exhibit blunted responses to IL-4 stimulation, indicating an impaired ability to adopt anti-inflammatory or tissue-repair-associated states [30,31]. Downregulation of CX3CR1–CX3CL1 and TGF-β1 signaling further compromises their ability to restrain inflammation [19,32]. Thus, aging promotes both inflammatory hyper-reactivity and loss of regulatory control, resulting in a self-sustaining state of chronic neuroinflammation.

This persistent inflammatory milieu adversely affects neuronal function. Pro-inflammatory cytokines impair neurotransmission and inhibit long-term potentiation (LTP), a fundamental memory-forming process in which synapses are strengthened through repeated stimulation. They also disrupt glutamate receptor trafficking and promote excitotoxicity and excessive synaptic pruning [33,34,35]. Collectively, these inflammation-related alterations diminish neural plasticity and contribute to age-associated learning and memory impairment [36].

### 2.4. Phagocytosis

Microglia play a central role in clearing cellular debris and misfolded proteins, but this phagocytic capacity declines markedly with age. In young brains, microglia efficiently engulf synaptic elements, apoptotic cells, and Aβ, thereby supporting tissue homeostasis. In contrast, microglia in the aged brain show reduced recognition and internalization of extracellular debris; for example, phagocytosis of myelin fragments is diminished in older mice, resulting in delayed or incomplete remyelination [37,38]. Aged microglia also exhibit impaired lysosomal function and accumulate undigested materials, including lipofuscin granules and lipid droplets [39,40]. These lipid droplet-accumulating microglia (LDAM) represent a dysfunctional state associated with impaired phagocytosis and heightened inflammatory cytokine release. Conversely, certain phagocytic pathways become aberrantly overactivated during aging. Complement-dependent phagocytosis, in particular, is enhanced and may drive excessive elimination of otherwise healthy synapses, a process known as excessive synaptic pruning [35,41,42].

Overall, microglia in the aged brain display impaired clearance and degradative capacity, promoting the accumulation of myelin debris, misfolded proteins, and Aβ, and thereby contributing to age-related neurodegenerative processes.

### 2.5. Frailty as a Therapeutic Window for Microglia-Targeted Intervention

Having outlined microglia as “foes,” we now propose that these adverse alterations already emerge during frailty, a prodromal stage preceding overt neurodegeneration. Crucially, frailty retains substantial plasticity, positioning this period as an optimal window for microglia-targeted preventive interventions. Cognitive frailty increases future dementia risk by nearly four-fold (HR = 3.99) [43], yet randomized controlled trials consistently show that resistance training [44] and multimodal programs combining aerobic exercise, strength and balance training, flexibility exercises, nutritional guidance, and cognitive training [45,46] can improve both physical and cognitive function within months. At the molecular level, whereas neurodegeneration involves irreversible processes such as neuronal loss, synaptic degeneration, and protein aggregation [47,48,49], frailty is characterized by milder and potentially reversible disturbances, including elevated inflammatory cytokines [50,51], accumulation of cellular remnants [52,53], lipid droplets [40], and incompletely degraded proteins resulting from reduced degradative capacity [54,55]. Because microglia orchestrate these inflammatory and clearance functions, modulating microglial activity during frailty may prevent the transition to neurodegeneration.

The senescence-accelerated mouse prone 8 (SAMP8) model recapitulates key features of cognitive frailty. SAMP8 mice exhibit muscle weakness and mild memory impairment at 4–6 months of age [56,57], along with chronic hippocampal inflammation, reduced neurogenesis, and age-dependent spongiform degeneration and atrophy [58]. Microglial function shifts markedly with age: at six months, microglia display exaggerated inflammatory responses to peripheral LPS, whereas by twelve months they become hyporesponsive and immunosuppressed [59]. These changes occur independently of amyloid and tau pathology, indicating that aging alone disrupts microglial homeostasis and contributes to cognitive decline. Physical frailty factors can further aggravate microglial dysfunction; for example, tooth loss exacerbates memory impairment in SAMP8 via cGAS–STING-dependent microglial pyroptosis [60], a process increasingly recognized as a driver of frailty-related neurodegeneration [61].

Pharmacological studies confirm that microglial dysfunction during early frailty remains reversible. Oral administration of Kai-Xin-San (KXS, 0.58 g/kg/day for 8 weeks) suppresses NLRP3 inflammasome activation in SAMP8 mice, reducing NLRP3, ASC, Caspase-1, GSDMD, and IL-1β/IL-18 levels and improving cognitive performance [62]; these effects are abolished by Nigericin, indicating dependence on the NLRP3/Caspase-1 pathway [62]. Taurine similarly ameliorates aberrant microglial reactivity and cognitive deficits, partly via upregulating TREM2 [63]. Together, these findings demonstrate that microglial plasticity is preserved during the early frailty phase.

However, this reversibility has clear limits. At four months, interventions attenuate inflammatory signaling and improve memory in SAMP8 [64], whereas by twelve months, microglia have transitioned into an immunosuppressed, “exhausted” state with markedly reduced therapeutic responsiveness [59]. In humans, 18–28% of individuals with mild cognitive impairment (MCI) revert to normal cognition [65,66]. However, hippocampal atrophy, which reflects neuronal loss, progresses more rapidly in MCI individuals who convert to Alzheimer’s disease (AD) than in non-converters [67,68,69]. Most clinical interventions slow atrophy rather than restore lost volume [70], and synaptic loss, which correlates strongly with cognitive decline, represents a major barrier to functional recovery [71,72].

Taken together, these observations suggest that the early stage of cognitive frailty, characterized by the coexistence of physical frailty and mild cognitive impairment, constitutes the optimal therapeutic window to leverage microglial plasticity and prevent progression toward dementia [69,73]. Nevertheless, the boundary between reversible and irreversible stages remains unclear, underscoring the need for future studies integrating microglial states with brain volume, neuronal and synaptic density, and cognitive trajectories to more precisely define the “point of no return” in the frailty–neurodegeneration continuum.

## 3. Aging Microglia as Friends

Thus far, we have focused on the detrimental aspects of microglial function that act as “foes” in brain aging and frailty. However, microglia do not solely function as neurotoxic or disease-promoting cells; under specific conditions, distinct subsets emerge that exert neuroprotective functions. These “friend-like” microglia play critical roles in debris clearance, synaptic support, and neuroprotection during aging and neurodegeneration (Figure 2, Table 1). In this chapter, we focus on several microglial subpopulations that emerge in response to aging and neurodegenerative stress: white matter-associated microglia (WAMs), which arise in response to myelin metabolic stress and degeneration during aging; disease-associated microglia (DAMs), which appear as aging progresses into neurodegenerative disease; activated response microglia (ARMs), which increase in the cortex and hippocampus along with amyloid pathology; and microglial neurodegenerative phenotype (MGnD), which represents a TREM2–APOE axis-dependent shift from homeostatic to neurodegenerative response states. Furthermore, we highlight CD11c^+^ microglia, which emerge during demyelination, white matter injury, and reparative phases of aging, and may represent a shared signature among protective microglial phenotypes. These microglial subsets exhibit upregulation of genes involved in phagocytosis, degradation, and tissue remodeling, suggesting that they act as “friends” that defend against age-related and neurodegenerative stress. However, under certain pathological or chronic inflammatory conditions, even these protective populations may transform into neurotoxic or inflammation-promoting states. This section summarizes their dynamics, gene expression programs, and functional characteristics.

### 3.1. White Matter-Associated Microglia (WAMs)

With advancing age, a distinct subset of microglia termed white matter-associated microglia (WAMs) emerges in the brain [77]. These cells are scarcely detected in young adult mice but accumulate prominently in white matter regions, such as the corpus callosum and medulla, in aged individuals [77]. Safaiyan et al. [77] identified WAMs as an aging-specific, TREM2-dependent microglial subtype that clusters around sites of myelin degeneration and participates in the clearance of myelin debris [77]. Transcriptomic analyses have revealed that WAMs upregulate genes involved in phagocytosis, lysosomal activity, and lipid metabolism, partially overlapping with the signature of DAMs [77]. Elevated expression of *Lgals3*, *Clec7a*, *Cd68*, *Itgax*, and *Spp1* indicates enhanced phagolysosomal and tissue-remodeling capacities, while increased *Apoe* and *Lpl* expression suggests activation of a lipid metabolic circuit that facilitates myelin lipid turnover [77]. Collectively, these molecular profiles suggest that WAMs contribute to myelin clearance and lipid recycling to preserve white matter homeostasis during aging.

Although generally considered protective, WAMs may acquire deleterious properties under conditions of pathological lipid accumulation. In aged mice, Galectin-3, a hallmark of WAMs, exacerbates neuronal injury in oxidized phosphatidylcholine (OxPC)–induced lesions, whereas Lgals3-deficient mice show markedly reduced damage [96]. This finding implies that WAMs can shift toward a pathological phenotype when lipid metabolism is impaired. Notably, similar upregulation of Galectin-3 has been observed in demyelinating lesions of patients with multiple sclerosis, suggesting the presence of WAMs-like microglia in the human brain [96]. Further studies using human post-mortem tissues and single-cell transcriptomic approaches will be essential to define how WAMs balance protective versus pathogenic roles in aging and demyelinating diseases.

### 3.2. Disease-Associated Microglia: DAMs

Disease-associated microglia (DAMs) represent a subset of microglia that emerge in response to the progression of neurodegenerative diseases, including AD. In AD mouse models, microglia begin to cluster around amyloid plaques at the early stage of plaque formation and differentiate into DAMs [74]. Genes upregulated in DAMs include *Trem2*, *Apoe*, *Tyrobp* (TYROBP; also known as DAP12), *Lpl*, *Cst7* (cystatin-F), *Itgax* (CD11c), *Clec7a*, *Cd9*, and *Cd63*, while homeostatic markers such as *P2ry12*, *P2ry13*, *Cx3cr1*, and *Tmem119*, are markedly downregulated [74]. Thus, the defining features of DAMs are the loss of homeostatic signatures and the induction of genes related to damage response, phagocytosis, and lipid metabolism [74].

Functionally, DAMs exhibit enhanced phagocytic activity. Immunohistochemical analyses in both mouse and human AD brains have revealed granular accumulation of Aβ within DAMs [74]. In low-density plaque regions, only 1.8% of microglia contained internalized Aβ, whereas in high-density plaque areas, this proportion increased to 60.6%, with 95.8% of these cells expressing LPL [74]. Similarly, CD11c/IBA1 double-positive microglia increased from 6% in wild-type mice to 22.3% in AD mice, localizing primarily in the cortex but not in the cerebellum [74]. These findings collectively indicate that microglia acquiring phagocytic capacity around Aβ deposits represent the DAMs phenotype.

A key regulatory axis underlying this transition is the TREM2-dependent signaling pathway [74]. The progression from homeostatic microglia to DAMs occurs in two stages: in the first (DAM1, intermediate state), suppression of homeostatic genes (e.g., *P2ry12/13*, *Cx3cr1*) and induction of *Apoe*, *Tyrobp*, and *B2m* occur independently of TREM2 [74]. In the second stage (DAM2, terminal state), TREM2 signaling drives strong upregulation of phagocytosis- and lipid metabolism-related genes, including *Lpl* and *Cst7* [74]. In *Trem2*-deficient mice, this transition is incomplete, with microglia remaining in an intermediate state [74]. Krasemann et al. [76] further demonstrated that phagocytosis of apoptotic neurons (dNs) rapidly induces Apoe, switching microglia from a homeostatic to a DAMs-like (MGnD) phenotype [76]. TREM2 signaling drives this *Apoe* expression, and APOE-dependent reprogramming is essential for phenotype conversion [76]. Indeed, *Apoe*-deficient microglia fail to fully adopt the DAMs-like transcriptional profile upon phagocytosis of apoptotic neurons, confirming the central role of the TREM2–APOE axis in DAMs programming [76].

Microglia form a compact microglial barrier around amyloid plaques and suppress the outward diffusion of Aβ aggregates through plaque compaction (compression and densification) [97]. This compaction process limits the formation of neurotoxic protofibrillar Aβ42 “hotspots” in non-covered regions. Deficiency of TREM2 (Trem2^−/−^, Trem2^+/−^) or human pathogenic variants (e.g., TREM2 p.R47H) lead to reduced plaque coverage, impaired amyloid compaction, and exacerbated axonal dystrophy (axonal degeneration) [98,99]. These findings indicate that DAMs, which are induced in a TREM2-dependent manner, contribute not only to the phagocytic clearance of amyloid but also to its structural containment. Moreover, in the presence of Aβ, TREM2 suppresses tau propagation and associated neurotoxicity [100]. TREM2 also binds to complement component C1q, inhibiting classical complement activation and thereby limiting excessive synaptic pruning (C3/iC3b-dependent microglial phagocytosis), ultimately supporting synaptic protection [101].

Under certain pathological conditions, DAMs may contribute to disease exacerbation. For instance, chronic activation of TREM2 by an agonistic antibody (AL002a) in 5xFAD mice with tau seeding worsened tau pathology and neuritic degeneration when Aβ clearance was insufficient [102]. Conversely, in tauopathy models without Aβ pathology, Trem2 deficiency attenuated microglial inflammatory responses and reduced neurodegeneration, despite minimal changes in tau burden [103]. In summary, DAMs exhibit both protective and detrimental properties depending on the pathological stage and microenvironment. Therefore, therapeutic strategies aiming to harness DAMs must involve precise temporal and contextual regulation of their activation and signaling programs to achieve beneficial outcomes.

### 3.3. Microglial Neurodegenerative Phenotype: MGnD

MGnD represents a DAM-like transcriptional state that emerges across multiple models of neurodegeneration. This state is characterized by the downregulation of homeostatic signature genes (*P2ry12*, *Tmem119*, *Cx3cr1*, *Csf1r*, *Tgfbr1*, *Mef2a*, *Sall1*) and the induction of genes associated with immune activation, phagocytosis, and lipid metabolism (*Apoe*, *Tyrobp*, *B2m*, *Itgax*, *Clec7a*, *Spp1*, *Axl*, *Lgals3*) [76]. Unlike DAMs, which were initially defined in the context of amyloid pathology, MGnD arises not only in AD models such as APP/PS1 mice, but also in experimental autoimmune encephalomyelitis (EAE) and amyotrophic lateral sclerosis (ALS; SOD1 model), indicating that it constitutes a shared and pathology-independent response of microglia to neurodegenerative stress [76]. The induction of MGnD depends on the TREM2–APOE signaling axis and is triggered by the phagocytosis of apoptotic neurons, which drives the transition from a homeostatic to a neurodegenerative phenotype [76]. In EAE models, the expression of homeostatic genes fluctuates in parallel with disease exacerbation and remission, reflecting the dynamic rise and fall of MGnD populations [76]. In AD models, DAMs-related markers such as CLEC7A become strongly enriched near amyloid plaques, suggesting that MGnD represents a broader transcriptional program encompassing such plaque-associated microglial states [74]. Consistent with this, Krasemann and colleagues identified MGnD as a cross-disease microglial response that reflects a common molecular reprogramming pattern induced by neuronal degeneration [76].

In summary, MGnD and DAM share a core gene signature, yet they differ in their scope and context: MGnD represents a disease-transcending reactive program, while DAM (as originally defined in 2017) refers to a localized, amyloid-driven response.

### 3.4. Activated Response Microglia: ARMs

When compared with DAMs, ARMs share several signature genes (*Apoe*, *Itgax*, *Clec7a*, *Spp1*, *Cst7*), but exhibit distinct induction pathways, spatial localization, and functional roles [74,75]. DAMs are induced in a TREM2-dependent manner and form compact barriers around amyloid plaques, contributing to plaque containment (“compaction”) [74,97,98], whereas ARMs are regulated primarily through APOE-dependent mechanisms and display a broader reactive phenotype characterized by enhanced antigen presentation and lysosomal activity [75]. Thus, while DAMs represent a spatially restricted, amyloid-containment phenotype, ARMs exhibit a more diverse and expansive reactive profile, encompassing both protective and pro-inflammatory properties. Overall, ARMs can be regarded as a broader disease-associated microglial state, positioned along the continuum of age-associated reactive microglial responses that become amplified under amyloid pathology.

At the transcriptional level, ARMs are characterized by the upregulation of genes involved in phagocytosis, lipid metabolism, and tissue remodeling (*Apoe*, *Itgax*, *Clec7a*, *Spp1*, *Cst7*, *Axl*), along with antigen presentation and lysosomal function (*Cd74*, *H2-Aa*, *H2-Ab1*, *Ctsb*, *Ctsd*) [75]. Conversely, the expression of homeostatic markers (*P2ry12*, *Cx3cr1*, *Tmem119*, *Csf1r*, *Sall1*) is markedly reduced, reflecting a transcriptional shift from a surveillant to a stress-adapted state [75]. Functionally, ARMs cluster around amyloid plaques, where they are implicated in debris clearance, lipid processing, and antigen presentation, potentially contributing to lesion containment and local tissue remodeling. However, excessive activation of antigen presentation or inflammatory pathways may lead to chronic inflammation and exacerbate neurotoxicity, depending on disease stage and context. Interestingly, ARMs-like transcriptional changes are also observed, albeit weakly, in aged wild-type mice (18 months old) in the absence of amyloid pathology, and are amplified by Aβ deposition in the same direction [75]. Thus, ARMs may represent an age-associated physiological microglial response that becomes pathologically expanded and intensified under amyloid stress.

### 3.5. CD11c^+^ Microglia

CD11c (*Itgax*) has attracted increasing attention as a unifying marker that links multiple protective microglial subsets exhibiting potential neuroprotective properties across the continuum from aging to neurodegenerative disease. CD11c (integrin αX: ItgαX) forms a heterodimer with CD18 to constitute the complement receptor CR4, a surface molecule expressed in microglia [14]. CD11c^+^ microglia emerge during development, aging, and pathological conditions (Figure 3), and are characterized by functional states associated with phagocytosis, lipid metabolism, and tissue remodeling [14]. This CD11c^+^ subset has been observed across various biological contexts, including development, aging, and neurodegeneration [14]. During aging, microglia in the white matter exhibit marked phenotypic transitions, accompanied by increased expression of CD11c (*Itgax*) and other phagocytosis-associated molecules, such as CD68, HLA-DR, *Clec7a*, *CD36*, *Axl*, *Lgals3*, *Apoe*, and *Lpl* [104,105]. These features have been consistently observed in both the corpus callosum of aged mice and the middle-aged human white matter [104,105].

In disease conditions, CD11c^+^ microglia have been detected in Alzheimer’s disease (AD), multiple sclerosis, amyotrophic lateral sclerosis, and stroke [106,107,108]. In AD models, for instance, CD11c^+^ microglia accumulate around amyloid plaques [74,109]. The proportion of CD11c^+^ microglia increases from approximately 12% in aged wild-type mice (21 months) to about 52% in AD model mice, indicating a progressive rise from aging to disease conditions [75].

CD11c^+^ microglia are defined by the upregulation of a core 22-gene module proposed by Benmamar-Badel et al. [14] comprising *Ank, Anxa5*, *Aplp2*, *Atp1a3*, *Clec7a*, *Colec12*, *Csf1*, *Ephx1*, *Fabp5*, *Fam20c*, *Gm1673*, *Gpnmb*, *Hpse*, *Igf1*, *Itgax*, *Lilrb4*, *Lpl*, *Nceh1*, *Plaur*, *Pld3*, *Plin2*, and *Spp1* [14]. These genes are associated with lipid handling, motility, and phagocytic processes [14]. Conversely, homeostatic markers such as *P2ry12*, *Tmem119*, *Sall1*, *Fcrls*, and *Cx3cr1* are downregulated, reflecting a transition toward a repair-oriented and phagocytic phenotype that resembles developmental or disease-associated states [14]. Among these, *Igf1* encodes insulin-like growth factor 1, a key neuroprotective molecule [110]. During development, CD11c^+^ microglia serve as the principal source of IGF-1 in the brain, and loss of *Igf1* leads to impaired primary myelination [110].

Although CD11c^+^ microglia possess a distinct gene signature, they share partial transcriptional overlap with other responsive subsets, including WAMs, DAMs, MGnD, and ARMs (Figure 2) [14,74,75,76,77]. As summarized in Table 1, the signature genes emphasized in each subset differ, yet three genes, namely *Itgax* (CD11c), *Apoe*, and *Clec7a*, are consistently upregulated across all subsets [14,74,75,76,77]. These molecules constitute key nodes in lipid metabolism and phagocytic pathways, underscoring their functional convergence. Notably, the homeostatic marker *P2ry12* is downregulated in all five subsets, indicating a common shift from homeostatic to responsive states dominated by lipid and phagocytic programs [14,74,75,76,77]. Collectively, CD11c can be regarded as a hallmark marker shared by neuroprotective microglial subsets, potentially reflecting a conserved mechanism of adaptive response. However, in aged white matter, genes related to phagocytic and response-associated processes, including *Lgals3* (Mac-2/Galectin-3), Fcγ receptor genes (e.g., *Fcgr1*, *Fcgr2b*, *Fcgr3*), *Clec7a* (Dectin-1), and *Cd36*, are also elevated, suggesting that CD11c^+^ microglia may enter a persistent reactive state under certain conditions [105]. Therefore, the interpretation of CD11c^+^ microglial function must consider disease stage, spatial localization, and microenvironmental context, as their role may vary from protective to maladaptive depending on the surrounding milieu.

**Figure 3 ijms-26-11494-f003:**
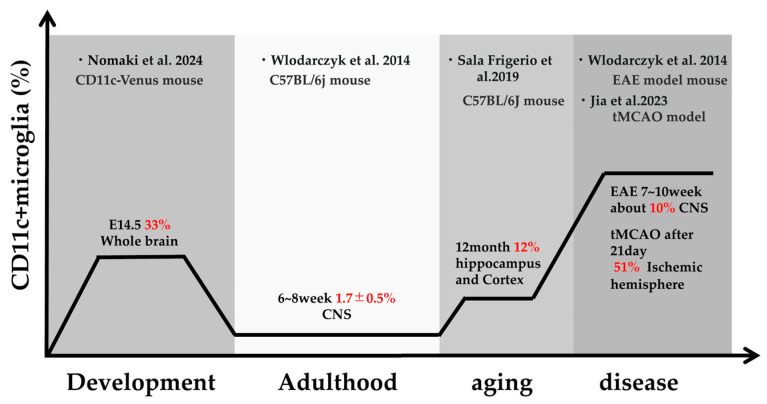
Schematic representation of CD11c^+^ microglia proportions. CD11c^+^ microglial ratios at each stage were integrated from studies using different mouse models and developmental or disease time points [75,80,111,112]. For the aged stage, the reference is the proportion of ARMs (*Itgax*-high subset) among total microglia.

## 4. Protective Microglial Subsets in Aging and Neurodegeneration

In the previous section, we summarized microglial subpopulations that may exert protective or adaptive functions during aging and age-related diseases. Among these, CD11c^+^ microglia represent one of the earliest reported and most well-characterized populations, regarded as a prototypical form of protective microglia that emerge during aging and neurodegenerative processes. Itgax (CD11c) is a core gene commonly upregulated across the protective subtypes described in the previous section and is widely used as an important marker for their identification and classification. In contrast, DAMs are a more recently identified subtype, for which research has rapidly advanced in recent years, elucidating the mechanisms of induction and transcriptional regulation in detail. DAMs accumulate around neurodegenerative lesions and share many molecular features with CD11c^+^ microglia. In this section, we focus on CD11c^+^ microglia and DAMs, providing an overview of their dynamics and functional significance in aging and neurodegenerative diseases (Table 2).

### 4.1. Alzheimer’s Disease

AD is the leading cause of dementia, characterized by the progressive decline of memory, cognition, and daily living abilities [122]. In recent years, increasing attention has been paid to disease-associated microglial subpopulations that emerge in response to AD pathology [74,109]. Among them, DAMs represent a major subset that is transcriptionally reprogrammed through activation of the TREM2–APOE signaling axis, leading to the induction of genes related to Aβ phagocytosis, inflammatory modulation, and neuroprotection (e.g., Apoe, Trem2, Itgax, Lpl, Cst7) [74,76]. Most DAMs correspond to CD11c^+^ microglia, which accumulate around amyloid plaques and increase in number as the disease progresses [109]. These cells actively engulf Aβ deposits, and their expression of the anti-inflammatory cytokine IL-10 is approximately 24-fold higher than that of CD11c^−^ microglia, suggesting a protective role through enhanced Aβ clearance and suppression of excessive inflammation [109]. The TREM2–APOE axis underlies this DAMs phenotype [74,99]. APOE plays a critical role in Aβ binding and phagocytic efficiency; notably, Apoe expression is elevated by approximately 22-fold in CD11c^+^ microglia derived from APP/PS1 mice compared with wild-type controls [109,123]. In humans, the *APOE* gene exists in three allelic variants (*ε2*, *ε3*, and *ε4*), where *ε2* is protective, while *ε4* is a well-established genetic risk factor for AD [122,123]. In 5xFAD mice, loss of TREM2 results in a near-complete absence of proliferative (Ki-67^+^) microglia around plaques, accompanied by reduced Aβ phagocytosis and altered plaque morphology with decreased compactness and modified Aβ subtype composition [99]. These findings indicate that TREM2 functions protectively in the context of Aβ pathology [99]. Clinically, elevated levels of soluble TREM2 (sTREM2) in cerebrospinal fluid are associated with attenuated cognitive decline and reduced neurodegenerative risk, particularly among *APOE ε4* carriers at the mild cognitive impairment (MCI) stage [124]. Collectively, these results suggest that activation of the TREM2–APOE pathway represents an intrinsic protective response to Aβ pathology.

On the other hand, functional heterogeneity exists within the DAMs population [74,113,114]. For example, Spp1 (osteopontin; OPN) is selectively expressed in CD11c^+^ microglia, but its role appears dualistic [113,114]. Secreted OPN contributes to the stabilization of CD11c expression, whereas OPN^+^ CD11c^+^ microglia release pro-inflammatory cytokines that exacerbate disease pathology [113,114]. Notably, treatment with an anti-OPN monoclonal antibody suppresses microglial inflammatory activation and ameliorates Aβ plaque pathology [114]. Moreover, during chronic Aβ phagocytosis, CD11c^+^ microglia accumulate lipid droplets (LDs), resulting in impaired phagocytic activity [115]. Inhibition of LD accumulation restores Aβ uptake, highlighting the importance of lipid metabolic homeostasis for sustaining microglial function [115]. Taken together, elucidating the diversity within neuroprotective microglial subsets and targeting their inflammatory and lipid metabolic pathways may provide novel therapeutic avenues for Alzheimer’s disease.

### 4.2. Amyotrophic Lateral Sclerosis (ALS)

ALS is a neurodegenerative disorder characterized by the selective loss of motor neurons, in which aging represents a major risk factor [125]. Microglial responses in ALS are highly heterogeneous, and recent studies have identified CD11c^+^ microglia and DAMs as two key subsets positioned at the center of the neuroprotective response in this disease. In ALS mouse models recapitulating TDP-43 pathology, a marked increase in CD11c^+^ microglia has been observed [108]. These cells display enhanced phagocytic activity toward TDP-43 aggregates, and their emergence has been associated with the suppression of phosphorylated TDP-43 (p-TDP-43) accumulation, attenuation of neuronal loss, and mitigation of motor deficits [108]. These findings suggest that CD11c^+^ microglia contribute to a protective compensatory response within the ALS-affected spinal cord.

Conversely, single-nucleus RNA sequencing analyses of transgenic mice overexpressing human mutant SOD1^G93A^ have identified the emergence of DAMs following motor neuron degeneration [116]. DAMs show a pronounced increase in the brainstem and spinal cord, progressively expanding in number as the disease advances and peaking during the late stage [116]. Two phases of accelerated motor neuron loss (P90–P110 and P130–P150) have been identified, during which the expression of several DAMs-associated genes fluctuates in synchrony with these degeneration phases [116]. Pseudotime trajectory analysis further revealed a continuous transcriptional transition from homeostatic microglia to the DAMs state, suggesting a gradual activation pathway rather than a discrete switch [116]. Importantly, DAMs-like profiles have also been detected in sporadic human ALS and *C9ORF72*-linked ALS, indicating that this response represents a conserved microglial program across species and etiologies [116]. DAMs and CD11c^+^ microglia share numerous molecular characteristics, and CD11c (*Itgax*) serves as a representative marker of the neuroprotective microglial subsets that include DAMs [116]. Indeed, CD11c is commonly used as a defining marker for DAMs in ALS models, and in vitro phagocytosis assays have shown that CD11c^+^ microglia engulf significantly more beads than other microglial subtypes [116]. Thus, CD11c^+^ microglia can be viewed as a functionally distinct subpopulation within the broader DAMs spectrum, characterized by enhanced phagocytic and tissue-repair activities.

Interestingly, in the lumbar spinal cord, particularly within regions controlling the hindlimbs where ALS symptoms appear earliest, a robust accumulation of CD11c^+^ microglia is accompanied by elevated expression of proinflammatory markers such as CD68, CCL2, IL-1β, and TNF-α [117]. These findings suggest that both CD11c^+^ microglia and DAMs exhibit dual and context-dependent roles, acting protectively in early or compensatory phases but potentially contributing to neuroinflammation and neuronal injury at later stages. Taken together, these data indicate that the roles of microglial subsets in ALS are phase-dependent and spatially heterogeneous. Therefore, effective therapeutic strategies should aim to delineate when and under which conditions these microglial populations should be promoted or suppressed, emphasizing the need for precise temporal control of microglial modulation in ALS.

### 4.3. Multiple Sclerosis (MS)

Multiple sclerosis (MS) is a chronic inflammatory disease of the central nervous system characterized by demyelination and neurodegeneration. Among the various factors influencing disease progression, age is the most critical determinant of clinical outcome [126]. In progressive forms of MS, age-related neurodegeneration and immunosenescence contribute to the accumulation of disability and a decline in therapeutic responsiveness, thereby exacerbating disease progression [127,128,129,130]. Moreover, several studies have reported telomere shortening and increased expression of aging-associated molecular markers in MS patients, suggesting an accelerated biological aging process [131]. Recent findings have highlighted that CD11c^+^ microglia play neuroprotective roles in experimental models of MS. In experimental autoimmune encephalomyelitis (EAE), an autoimmune model recapitulating MS-like inflammation, CD11c^+^ microglia exert protective effects against demyelination and disease progression [118]. Intrathecal administration of CSF1R ligands (CSF1 or IL-34) at the onset of disease increases the number of CD11c^+^ microglia, concomitantly reducing clinical severity and spinal cord demyelination [118]. CSF1R stimulation enhances *Ccl2* expression in the brain, and the expression levels of *Ccl2* and *Itgax* (CD11c) are positively correlated [118]. Overexpression of CCL2 further increases CD11c^+^ microglia in a CCR2-independent manner, accompanied by elevated *Csf1* mRNA expression [118]. These findings indicate that the CSF1R–CCL2 signaling axis promotes the induction of CD11c^+^ microglia, thereby contributing to neuroprotection in MS.

In a complementary approach, conditional deletion of *p38α* in CD11c^+^ cells resulted in a striking disease acceleration in approximately 60% of female EAE mice after day 30, coinciding with a marked reduction in CD11c^+^ microglia [119]. Interestingly, in non-progressing animals, CD11c^+^ microglia were found in close contact with astrocytic processes and the glia limitans, maintaining immune cells confined to perivascular spaces [119]. In contrast, animals exhibiting a loss of CD11c^+^ microglia displayed an expansion of CCL2-high reactive astrocytes and enhanced inflammatory infiltration into the white matter parenchyma [119]. These findings suggest that CD11c^+^ microglia contribute to the maintenance of the vascular–parenchymal barrier and suppression of inflammatory spread within the CNS.

Further evidence supporting the protective role of CD11c^+^ microglia comes from studies using *Sirpα*-deficient mice. SIRPα, a microglial surface receptor that interacts with its ligand CD47, negatively regulates the emergence of CD11c^+^ microglia [13]. Loss of *Sirpα* leads to a selective increase in CD11c^+^ microglia within white matter regions [13]. In the cuprizone-induced demyelination model, microglia-specific deletion of *Sirpα* significantly attenuated demyelination and reduced the area and density of Iba1^+^ cells [13]. These results indicate that the SIRPα–CD47 axis functions as a “brake” on the emergence of reparative CD11c^+^ microglia, and that lifting this inhibitory signal facilitates white matter repair [13]. Notably, the increase in CD11c^+^ microglia also occurs in the absence of demyelination, and their gene expression profile closely resembles that of the remyelination phase, characterized by elevated expression of *Itgax*, *Igf1*, *Spp1*, and *Lpl* [13]. Collectively, these studies identify CD11c^+^ microglia as central mediators of white matter repair and containment of inflammation in MS. The inductive CSF1R–CCL2 pathway and the disinhibitory SIRPα–CD47 pathway represent two complementary mechanisms regulating the emergence and function of these cells. A deeper understanding of these signaling networks may provide novel therapeutic avenues aimed at enhancing CD11c^+^ microglial responses to counteract age-related progression and promote remyelination in progressive MS.

### 4.4. Ischemic Stroke

The risk of stroke increases progressively with age, doubling approximately every decade after 45 years of age [132]. More than 70% of stroke cases occur in individuals over 65 years old, highlighting aging as the strongest risk factor for stroke onset [132]. Moreover, post-stroke functional recovery declines markedly with age. In patients undergoing endovascular thrombectomy (EVT), the rate of regaining near-independence at 90 days (modified Rankin Scale, mRS 0–2) is approximately 30% in individuals over 80 years old, compared with 50–55% in those under 80 [133]. Similarly, the proportion achieving ambulatory independence (mRS 0–3) drops from 56% in patients under 60 years to only 15% in those over 80 [134]. These data clearly indicate that stroke not only occurs more frequently in the elderly but also that recovery potential diminishes with advancing age.

Recent studies have demonstrated that CD11c^+^ microglia are prominently induced in the white matter during the recovery phase after ischemic stroke and play a central role in white matter repair. Jia et al. [80] showed in a mouse model of cerebral infarction that white matter injury peaks at 7 days post-stroke and gradually recovers by day 30, during which CD11c^+^ microglia accumulate in the lesion area [80]. These cells exhibit elevated expression of genes related to phagocytosis (e.g., *Axl*, *Cd68*), myelin-supportive functions (*Igf1*, *Spp1*, *Csf1*), and lipid metabolism (*Apoe*, *Apoc1*, *Abca1*), suggesting that they contribute to white matter regeneration by clearing myelin debris and promoting oligodendrocyte differentiation and maturation [80]. Selective depletion of CD11c^+^ microglia markedly impaired remyelination and significantly reduced both motor and cognitive recovery [80]. Collectively, these findings identify CD11c^+^ microglia as a key reparative subset driving white matter restoration and functional recovery after ischemic stroke.

In addition to this reparative response, DAMs-like microglial subsets also emerge during both the acute and chronic phases after stroke. Kim et al. [120] performed single-cell RNA sequencing in a transient middle cerebral artery occlusion (tMCAO) model and identified a population termed Stroke-Associated Microglia (SAM) that displays a DAMs-like transcriptional profile. SAMs upregulate antioxidant-related genes and are suggested to exert neuroprotective functions under ischemic stress [120]. Similarly, Cao et al. [121] reported a robust accumulation of CD11c^+^ microglia in the ipsilateral thalamus following cortical infarction, coinciding with progressive thalamic degeneration. By day 14 post-stroke, a significant increase in CD11c expression was observed, and by day 28, nearly all Iba1^+^ microglia in the affected thalamus expressed CD11c [121]. These cells displayed reduced expression of homeostatic markers (*Tmem119*, *Cx3cr1*) but strong upregulation of DAMs-associated genes such as *Apoe*, *Axl*, *Lpl*, *Cst7*, and *Csf1*, suggesting that they represent DAMs-like microglia involved in debris clearance and suppression of secondary neurodegeneration [121]. Together, these findings indicate that DAMs-like responses arise sequentially across acute and chronic phases, reflecting a dynamic and stage-specific transition of microglia toward a protective phenotype following ischemic injury.

Overall, CD11c^+^ and DAMs-like microglia constitute a family of repair-oriented microglial subtypes that orchestrate tissue restoration and functional recovery after stroke. Their appropriate induction and maintenance may compensate for the limited regenerative capacity of the aged brain and represent a promising target for enhancing long-term recovery in elderly stroke patients.

## 5. Mechanisms Regulating the Protective (Friend-like) Microglial State

In the previous section, we summarized current knowledge regarding microglial subpopulations that can act as protective allies in the context of aging and age-related neurodegenerative diseases. Enhancing the population of protective microglia, while preventing their transition into pathological phenotypes, is considered a critical therapeutic strategy for combating frailty and neurodegeneration. In this section, we aim to provide insight into the regulatory mechanisms underlying protective microglia by focusing on two key aspects: (1) the molecular and cellular switches that drive the emergence of CD11c^+^ microglia, and (2) the conditions under which microglia maintain their protective properties versus those that trigger a pathological shift.

### 5.1. Inductive Mechanisms of Protective Microglia

Although the conditions that drive the emergence of CD11c^+^ microglia have not been fully defined, several studies provide converging clues. Using *Itgax*-Venus reporter mice, Kohno and colleagues visualized CD11c^+^ microglia in the dorsal horn and showed that phagocytosis of myelin debris after peripheral nerve injury triggers CD11c induction [79]. Shen et al. [113] reported that efferocytosis of apoptotic neurons promotes differentiation from CD11c^−^ precursors to CD11c^+^ microglia, implicating efferocytosis itself as a primary switch for CD11c emergence [113]. In a cuprizone demyelination model, loss of *Cx3cr1* attenuated the appearance of CD11c^+^ microglia, indicating that CD11c induction depends on CX3CR1-mediated phagocytic activity [135]. Consistently, deletion of the “don’t-eat-me” SIRPα–CD47 axis increased CD11c^+^ microglia even without overt injury, suggesting that release of phagocytic brakes facilitates this program [13]. At the transcriptional level, ingestion of dead cells can reprogram microglia toward a neurodegenerative-response state (MGnD) [76]. Together, these findings support a model in which phagocytosis of lipid-rich cargo (especially myelin) and apoptotic material serves as a principal switch for the emergence of protective, CD11c-bearing microglia. Clinically and preclinically, white-matter changes are common in frail older adults, and very early myelin abnormalities are reported across aging-related disease models, suggesting that microglia may mount such debris-clearance programs prior to overt clinical symptoms as a defensive response.

The TREM2–APOE pathway, which is required for stage-2 DAMs, links these phagocytic cues to effector programs. TREM2 expression enhances uptake of apoptotic neurons [136], TREM2 overexpression enables efficient engulfment of apoptotic Neuro2A cells via injury-induced neuronal ligands [137], and TREM2 binds apolipoproteins (APOE, CLU/APOJ) to promote Aβ uptake [138]. This has been demonstrated in multiple models [76,77,83], positioning the TREM2–APOE axis as a central molecular switch that links phagocytic cues to the induction of protective microglial states. In TREM2-deficient mice, impaired DAMs activation can be partially rescued by supporting cellular energy metabolism [139], indicating a requirement for metabolic reprogramming. Downstream SYK signaling is essential for DAMs induction, and CLEC7A engagement can elicit DAMs-like states independently of TREM2 [140]. Moreover, simple agonism of TREM2 with antibodies can produce microglial states distinct from canonical DAMs, underscoring that phagocytic activity, lipid handling, and bioenergetic homeostasis operate as integrated regulators of protective microglial emergence rather than a single linear pathway [141].

In summary, available evidence supports a three-axis model for the induction of CD11c^+^/protective microglia: (1) Phagocytic triggers (myelin debris, apoptotic cells), (2) Receptor circuits (TREM2–APOE, CX3CR1, SIRPα–CD47, CLEC7A–SYK), and (3) Metabolic rewiring (energy support and lipid metabolism), which together bias microglia toward debris clearance, lipid processing, and tissue-protective functions early in the frailty–neurodegeneration continuum.

### 5.2. Balancing Lipid Metabolism Determines the Functional State of Microglia

The ability of microglia to metabolize and recycle lipids and cholesterol derived from phagocytosed myelin debris or apoptotic cells critically determines their functional phenotype. In demyelinating environments, microglia take up large amounts of myelin-derived cholesterol, and their capacity to process these lipids dictates whether they adopt a pro-inflammatory or reparative trajectory. Berghoff et al. [142] demonstrated, using multiple demyelination models, including cuprizone, EAE, and lysophosphatidylcholine, that cholesterol metabolism in microglia becomes optimized during the remyelination phase. During demyelination, the expression of sterol synthesis enzymes is broadly upregulated, whereas *Dhcr24*, the enzyme that catalyzes the reduction of desmosterol to cholesterol, is selectively downregulated. As a result, desmosterol accumulates intracellularly and acts as an endogenous ligand for liver X receptors (LXRs; *Nr1h2*/LXRβ and *Nr1h3*/LXRα). Activation of LXR induces the expression of cholesterol efflux transporters ABCA1 and ABCG1, as well as the lipid carrier APOE, facilitating efficient export and recycling of excess myelin-derived lipids. Simultaneously, LXR activation suppresses pro-inflammatory cytokine genes such as *Il1b* and *Tnf*, thereby promoting the resolution of inflammation. Through this coordinated metabolic reprogramming, microglia transition into a reparative metabolic state capable of coupling lipid handling with inflammatory restraint, thereby supporting oligodendrocyte differentiation and remyelination [142].

Conversely, conditional deletion of squalene synthase (SQS; *Fdft1*) in microglia impairs desmosterol accumulation and LXR activation, leading to insufficient induction of ABCA1, ABCG1, and APOE, while elevating inflammatory gene expression. Histologically, this is accompanied by enlarged MAC3^+^ microglial areas and reduced remyelination markers (Olig2^+^/CAII^+^ cells and MBP^+^ regions), indicating defective tissue repair. Thus, the pathway “*Dhcr24* downregulation → desmosterol accumulation → LXR activation” represents a central mechanism that synchronizes lipid clearance and inflammatory resolution to drive post-demyelination repair [142]. Collectively, these findings underscore that efficient cholesterol metabolism is essential for inflammation control and remyelination in microglia. Maintaining proper lipid catabolism and efflux capacity is therefore a key determinant of protective microglial phenotypes that support neural repair and homeostasis (Figure 4).

### 5.3. TREM2 Coordinates the “Digestion” and “Efflux” Phases of Microglial Debris Processing

Microglia process fragmented myelin and amyloid-associated debris through a sequential cascade of uptake, lysosomal digestion, and lipid efflux [77]. Among these steps, accumulating evidence indicates that TREM2 plays a particularly critical role in the second (digestion) and third (efflux) phases of this pathway [77,83]. In aging white matter, Safaiyan et al. [77] demonstrated that TREM2-dependent WAMs exhibit transcriptional activation of genes involved in phagocytosis and lipid metabolism, enabling the breakdown and clearance of myelin-derived lipids [77]. In contrast, *Trem2*-deficient mice fail to fully transition into this WAMs-like state, resulting in delayed degradation of myelin fragments [77]. Notably, debris uptake remains largely intact, while the lysosomal degradation phase is impaired, as reflected by the downregulation of cathepsins and other lysosomal enzymes, as well as an overall failure to induce the WAMs transcriptional program [77].

The consequence of this impaired degradation is tightly coupled to defective lipid efflux, as shown by Nugent et al. [83], who performed cell-type-specific lipidomics and transcriptomic profiling in a chronic demyelination (cuprizone) model [83]. In *Trem2*-deficient mice, microglia displayed a marked accumulation of cholesteryl esters (CEs) and oxidized CEs, indicating an inability to efficiently export cholesterol [83]. Importantly, this lipid buildup was specific to microglia [83]. Pharmacological inhibition of ACAT1, the enzyme that converts free cholesterol into CE, reduced lipid accumulation, while activation of the nuclear receptor LXR induced expression of cholesterol transporters (*Abca1*, *Abcg1*) and the lipid carrier *Apoe*, thereby normalizing CE levels [83]. These findings suggest that the NPC2 → LXR–ABCA1/ABCG1/APOE efflux axis fails to operate effectively in the absence of TREM2, creating a metabolic bottleneck that drives microglial foam cell–like lipid overload and increases the risk of neurotoxicity. A review by Damisah et al. [143] further conceptualized these studies within a unified framework, proposing that TREM2 orchestrates the entire continuum from lipid degradation to efflux [143]. From this perspective, pharmacological activation of LXR or inhibition of ACAT1 may help relieve the metabolic “traffic jam” caused by defective TREM2 signaling.

These findings indicate that TREM2 supports both lysosomal digestion and cholesterol efflux, sustaining the degradative and metabolic competence of microglia. In settings of TREM2 deficiency, whether genetic or age-associated, these two processes are simultaneously compromised, leading to lipid accumulation, chronic inflammation, and neurodegeneration. Therefore, future therapeutic approaches may need to target the coordinated enhancement of both digestion and efflux pathways, tailored to the disease stage or aging context.

### 5.4. Lipid Metabolic Dysfunction: Inflammation Driven by Lipid Droplets and Cholesterol Crystals

When lipid metabolism becomes dysregulated, excessive neutral lipids accumulate within the cytoplasm, leading to an expansion of lipid droplets (LDs). In the aged brain, a distinct population of lipid droplet-accumulating microglia (LDAM) emerges, characterized by reduced phagocytic capacity, elevated reactive oxygen species (ROS) production, lysosomal dysfunction, and excessive expression of proinflammatory cytokines [40]. Consequently, these microglia fail to effectively process Aβ or myelin debris, resulting in persistent accumulation of cellular remnants and a self-perpetuating cycle of chronic neuroinflammation [40].

In parallel, disruption of cholesterol homeostasis, either through dysfunction of the ABCA1/ABCG1–ApoE efflux pathway or through impaired esterification, leads to intracellular cholesterol supersaturation [144]. This promotes the formation of cholesterol crystals, which trigger activation of the NLRP3 inflammasome, thereby inducing the production of inflammatory cytokines such as IL-1β and impairing tissue repair and remyelination [144,145].

Importantly, the accumulation of LDs or cholesterol crystals is not the primary cause of microglial dysfunction but rather a secondary manifestation of defective lipid handling. In *Trem2*-deficient mice, microglia exhibit pronounced accumulation of cholesteryl esters, a phenotype that can be ameliorated by ACAT1 inhibition or LXR activation, both of which enhance the re-esterification and efflux of free cholesterol [143]. Similarly, in aged or demyelinated brains, pharmacological activation of LXR upregulates the *Apoe*–*Abca1*–*Abcg1* axis, suppressing cholesterol crystallization and chronic inflammation while promoting white matter repair [144]. Furthermore, in recent AD models, depletion of the lipid droplet biogenesis factor *fit2* reduces LD burden, enhances CD11c^+^ microglial phagocytosis of Aβ, and significantly decreases both Aβ deposition and inflammatory marker expression in the brain [115].

Collectively, these findings underscore that maintaining a balanced cycle of lipid digestion and efflux is essential for sustaining the protective microglial phenotype. Efficient lipid processing prevents the buildup of lipid droplets and cholesterol crystals, thereby safeguarding microglial homeostasis and supporting tissue repair in aging and neurodegenerative conditions [146].

### 5.5. Microglial States Determine the Efficacy of Aβ Clearance by Immunotherapy

Microglia play a pivotal role in the mechanism of Aβ clearance induced by the emerging Aβ immunotherapies for Alzheimer’s disease. Although these therapies target Aβ directly, the actual removal of Aβ is largely executed by microglia rather than the antibody itself. Postmortem analyses of patients who received active Aβ immunization (AN1792, an Aβ_42_ peptide vaccine that induces anti-Aβ antibodies) revealed that the extent of Aβ clearance strongly correlated with the functional state of microglia [147]. Specifically, in cases showing substantial Aβ reduction, microglia exhibited suppressed production of proinflammatory cytokines together with enhanced oxidative phosphorylation and lipid metabolic activity. Conversely, in patients with limited Aβ removal, persistent activation of the IL-2–STAT5 pathway and complement cascade was observed, indicating that their microglia remained in a chronic inflammatory state. These findings collectively suggest that optimizing microglial metabolic pathways while restraining inflammatory signaling may be a key strategy to enhance the efficacy of Aβ clearance in immunotherapy [147].

## 6. Taking Microglia to the Clinic: Next Moves Informed by TREM2 Setbacks

Recently, an agonistic antibody targeting TREM2 (AL002; INVOKE-2 trial) was evaluated in patients with Alzheimer’s disease to test the therapeutic potential of modulating microglial function through the TREM2 pathway. However, this phase 2 study failed to meet its primary endpoint of cognitive improvement, and changes in Aβ clearance and neurodegeneration biomarkers were limited [148] in a subset of patients, the INVOKE-2 trial did not show significant improvements in neuroinflammatory biomarkers, amyloid reduction, or clinical outcomes, suggesting that TREM2 pathway modulation alone may be insufficient to achieve neuroprotection [148,149]. First, these findings indicate that stimulation of the TREM2 pathway alone may not be sufficient to induce or maintain a stable neuroprotective microglial phenotype. Although TREM2 is a key regulator of phagocytic initiation and lipid handling, it must be accompanied by proper inflammatory control; otherwise, the transcriptional state of microglia may shift toward a pro-inflammatory direction. Indeed, SPP1 (osteopontin) is frequently expressed in CD11c^+^ microglia but does not necessarily represent a protective signature. Among CD11c^+^ microglia, SPP1^+^ cells show higher expression of inflammatory cytokines (e.g., TNF-α) and reduced Aβ uptake, whereas SPP1^−^ cells exhibit elevated expression of TREM2, Axl, and MerTK, along with enhanced lysosomal activity and efficient Aβ degradation [114]. This functional bifurcation, in which microglia diverge into inflammatory and reparative states even under the same TREM2 pathway, suggests that TREM2 signaling alone cannot stabilize the desired phenotype. Hence, the INVOKE-2 results underscore the importance of combined strategies that simultaneously ensure the integrity of the “uptake–digestion–clearance” metabolic cycle and maintain appropriate inflammatory regulation.

Second, patient stratification analyses suggest that the timing of intervention may critically determine therapeutic responsiveness. During the pre-MCI to early MCI phase, microglial metabolism and phagocytic capacity remain adaptable and potentially recoverable through environmental or therapeutic modulation. In contrast, in later disease stages, impaired lipid efflux and self-sustaining inflammatory loops become functionally locked, reducing responsiveness to TREM2-based interventions. Therefore, the transition from frailty to MCI should be regarded as a “window of reversibility”, during which it is most rational to examine whether microglial functions can be restored or guided toward protective states. To define the limits of reversibility, systematic analyses during the cognitive frailty stage are required, supported by objective multimodal biomarkers integrating fluid, imaging, and spatial transcriptomic data that reflect changes in inflammation, metabolism, and phagocytic function.

## 7. Conclusions

Whether microglia are friend or foe—that is the question. This question has guided our exploration throughout this review. Microglia embody both destruction and repair, inflammation and resolution. Their fate, and ours, depends on which side we choose to nurture. It remains for us to answer through future research. To approach this question scientifically, we have synthesized current evidence on the dual nature of microglia in brain aging and age-related disease, cells capable of acting as both “friends” and “foes.” On the one hand, microglia can contribute to disease onset and progression through impaired surveillance and chronic inflammation. On the other hand, subsets of protective microglia resist pathology through robust phagocytic capacity. Notably, even these beneficial cells can switch to a disease-promoting phenotype when lipid metabolism fails. In essence, a continuously efficient “phagocytosis–metabolism–efflux” axis, coupled with restraint of runaway inflammation, defines the protective microglial state, and maintaining this homeostasis is likely critical for countering frailty and age-related disorders.

A priority for future research is to elucidate the induction and expansion mechanisms of “friend-like” microglia, including CD11c^+^ protective subsets. The stimuli, dosing, and temporal patterns that drive the emergence and amplification of this phenotype remain insufficiently defined. Beyond gene-expression profiling, we need time-resolved evidence that tracks functional microglial outputs in the aging brain, including phagocytosis, lipid handling, remyelination, and synapse preservation, and links these to behavioral outcomes (i.e., mitigation of cognitive decline). Importantly, frailty represents a pre-disease, potentially reversible window; therefore, mechanistic studies in this stage are most likely to translate into clinical benefit. Dissecting how friend-like microglia are maintained or induced during frailty and determining their impact on disease containment and healthy aging should be considered a top priority.

A key challenge for future research is to define the “reversible window” along the frailty-to-disease continuum, clarifying how long functional reversibility persists and when therapeutic intervention is most effective, using well-defined histological and functional biomarkers. However, preventive trials initiated at truly preclinical or frailty stages remain limited, and the lack of standardized criteria hampers translation. Furthermore, regional heterogeneity (e.g., hippocampus vs. neocortex) and species differences (human vs. mouse) strongly influence microglial states and therapeutic responsiveness, emphasizing the need for integrated and comparative study designs across models and brain regions. The limitations of single-target strategies are also becoming apparent. For example, purely activating TREM2 may be insufficient; instead, combined approaches that optimize lipid metabolism while damping inflammation will likely be required. Yet preclinical and clinical evidence for such combination strategies remains limited. From a translational standpoint, techniques such as engrafting human iPSC-derived microglia into mouse brains are becoming feasible [150], but rigorous testing focused on protective microglial states is still lacking. Human-specific analyses using iPSC-microglia could provide essential momentum toward clinical application.

By progressively closing these gaps, we may learn to stabilize microglia in their protective “friend” state and harness them for preventive or early disease-modifying interventions. Even microglia acting as foes may, in the future, be converted into friends through the development of novel therapeutic agents. Current work is still weighted toward transcriptional analyses; integrative studies that bridge cellular function to circuit dynamics and, ultimately, to behavior and cognition are urgently needed. Advancing such multilevel approaches will be crucial to establishing microglia as clinically actionable targets and to realizing their full potential in maintaining brain health across aging.

## Figures and Tables

**Figure 1 ijms-26-11494-f001:**
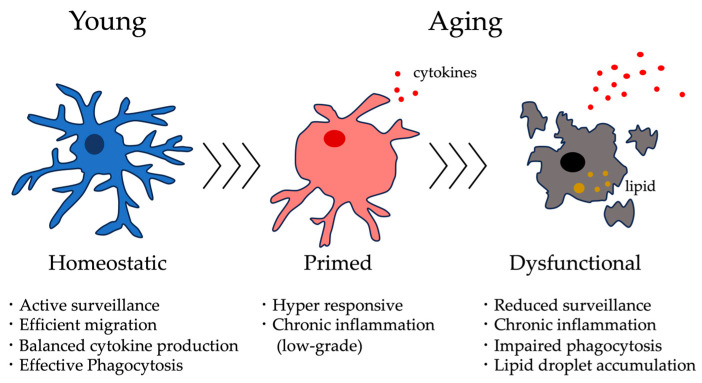
Aging microglia as foes. During aging, microglia shift from a homeostatic state with active surveillance and efficient phagocytosis to a primed, proinflammatory phenotype, and finally to a dysfunctional state characterized by impaired clearance, chronic inflammation, and lipid droplet accumulation.

**Figure 2 ijms-26-11494-f002:**
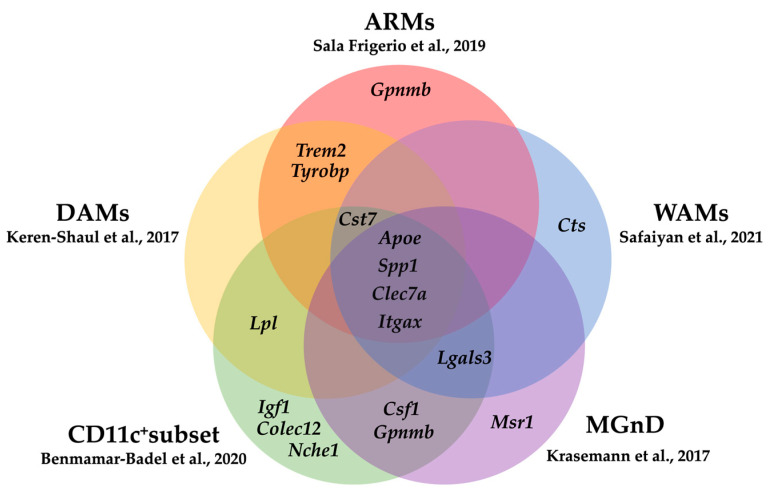
Overlapping gene signatures among microglial subsets. Friend-like protective microglial populations such as DAMs [74], ARMs [75], MGnD [76], WAMs [77], and CD11c^+^ microglia [14] share core genes (*Apoe*, *Spp1*, *Clec7a*, *Itgax*) related to phagocytosis and lipid metabolism.

**Figure 4 ijms-26-11494-f004:**
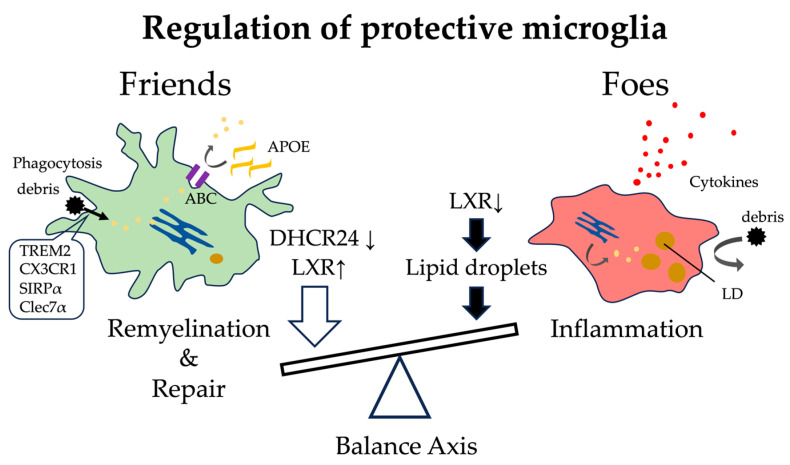
Regulation of friends-like microglia. Microglia maintain a balance between debris clearance and inflammation. LXR activation induced by DHCR24 downregulation promotes APOE-mediated lipid efflux and remyelination (“friends”), whereas impaired LXR signaling drives lipid droplet accumulation and cytokine release (“foes”). Arrows next to molecules indicate upregulation or activation (↑) and downregulation or impairment (↓). The block arrows pointing to the balance beam indicate the factors driving the shift in microglial phenotype.

**Table 1 ijms-26-11494-t001:** Upregulated genes across the five microglial subsets (WAMs, DAMs, MGnD, ARMs and CD11c^+^ microglia).

Gene	Protein		References
**Commonly upregulated across all five subsets**			
*Apoe*	ApoE	Regulates lipid transport and Aβ uptake	[78]
*Itgax*	CD11c (αX integrin)	α-subunit of complement receptor 4 (CR4), noted as a marker of the subset that supplies IGF-1	[79,80]
*Clec7a*	Dectin-1	Induces inflammatory and phagocytic responses	[81]
*Spp1*	Osteopontin (OPN)	Promotes phagocytosis and tissue repair, but can also amplify chronic inflammation	[82]
**Upregulated genes, including potential “friends” of microglia**			
*Trem2*	TREM2	Recognizes lipids, Aβ, apoptotic cells; regulates lysosomal function and cholesterol efflux	[83]
*Lpl*	Lipoprotein lipase	Controls lipid debris clearance and supports tissue repair/remyelination	[84]
*Igf1*	Insulin-like growth factor-1	Contributes to neuroprotection and myelin repair/remyelination	[85]
*Nceh1*	Neutral cholesterol ester hydrolase 1	Hydrolyzes cholesterol esters, promotes efflux, suppresses foam cell formation	[86]
*Msr1*	Scavenger receptor A1	Clears degenerating myelin and Aβ-related debris	[87]
*Gpnmb*	GPNMB	Enhances phagocytic and lysosomal activity, promotes reparative responses	[88]
*Csf1*	CSF1 (ligand for CSF1R)	Maintains microglial survival and homeostasis via CSF1R signaling	[89]
*Axl*	AXL (TAM RTK)	Recognizes apoptotic cells and Aβ to trigger phagocytosis	[90]
*Colec12*	Collectin-12 (CL-P1/SRCL)	Facilitates myelin and debris uptake for white matter repair	[91]
*Lgals3*	Galectin-3	Enhances phagocytosis and remyelination; may also amplify inflammation	[92]
*Tyrobp*	DAP12 (TYROBP)	Supports protective microglial responses; excessive ITAM signaling may drive inflammation	[93]
*Cst7*	Cystatin F	Regulates endolysosomal trafficking and function	[94]
*Cts*	Cathepsin	Degrade cellular debris and enable Aβ fibril degradation via lysosomal acidification	[95]

**Table 2 ijms-26-11494-t002:** Summary of protective microglial subsets across neurological diseases.

Disease	Subset	Protective Function	Detrimental Function	Mechanisms	References
Alzheimer’s disease (AD)	DAMs CD11c^+^ microglia	Early stage Aβ clearance ↑ Inflammatory ↓	Late stage Phagocytosis ↓ Inflammation ↑	TREM2–APOE IL10 lipid-metabolic	[74,76,99,109,113,114,115]
Amyotrophic lateral sclerosis (ALS)	DAMs-like CD11c^+^ microglia	Early stage TDP-43 clearance ↑ neuronal loss ↓	Late stage Inflammation ↑	Phagocytosis Inflammatory-cytokines	[108,116,117]
Multiple sclerosis (MS)	CD11c^+^ microglia	Demyelination ↓ Inflammation ↓ Remyelination ↑ White-matter repair ↑	—	CSF1R- CCL2 SIRPα–CD47	[13,118,119]
stroke	CD11c^+^ microglia SAM (DAMs-like)	White-matter repair ↑ Remyelination ↑ Myelin debris clearance ↑	—	Phagocytosis IGF-1 lipid-metabolic	[80,120,121]

The upward arrow (↑) indicates an increase or upregulation, and the downward arrow (↓) indicates a decrease or downregulation.

## Data Availability

No new data were created or analyzed in this study. Data sharing is not applicable to this article.

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
