# Peer review of "Microglia in Brain Aging and Age-Related Diseases: Friends or Foes?"

_ijms, 2025, doi:10.3390/ijms262311494_

Round 1
Reviewer 1 Report
Comments and Suggestions for Authors
The review provides a clear and comprehensive overview of microglial populations and their associated markers. It addresses an important and timely topic, particularly given the limitations of the traditional M1/M2 classification paradigm. Overall, the manuscript is interesting and relevant, but several aspects require refinement before publication.
Minor revision.
1 The organization of Section 2.1 (“Surveillance Function (Environmental Monitoring and Process Motility)”) should be improved to ensure a more coherent narrative flow. The text alternates between human and mouse microglia in a way that feels fragmented.
2 A central strength of the review is its effort to move beyond the oversimplified M1/M2 framework. However, the manuscript still employs this terminology in a way that may appear contradictory to its main message. For example: “microglia in aged brains exhibit blunted responses to IL-4 stimulation, reflecting impaired polarization toward the M2 phenotype [22,23].”
3 The subsection “2. Aging Microglia as Foes” is comparatively long and, in its current form, largely reiterates information that is already well established in the literature and covered in previous reviews. Considering that the core contribution of this article is the characterization of microglial subsets and state-specific changes. Authors should condense this section to avoid redundancy and to maintain focus on what is novel or particularly integrative in the present review.
4 The manuscript would be substantially strengthened by the inclusion of a summary table compiling key information for each disease or condition discussed.
5 The use of dashes can add clarity and elegance to scientific writing when used to create emphasis or refine sentence rhythm. However, in the current version, dashes are employed excessively, which at times disrupts the flow and lends an overly informal tone to the text.
Author Response
We sincerely thank the reviewer for their thoughtful and constructive comments. We have carefully revised the manuscript in accordance with all suggestions. Below, we provide a point-by-point response. All changes have been highlighted in the revised manuscript.
Comment 1: The organization of Section 2.1 (“Surveillance Function (Environmental Monitoring and Process Motility)”) should be improved to ensure a more coherent narrative flow. The text alternates between human and mouse microglia in a way that feels fragmented.
Response 1: Thank you for this valuable suggestion. We have reorganized Section 2.1 to improve the narrative flow (p3. Low 94-103). Specifically, we now (1) introduce the general principles of microglial surveillance, (2) summarize findings from mouse studies, and (3) present corresponding observations in the human brain. In addition, we simplified the description to avoid unnecessary repetition.
Comment 2: A central strength of the review is its effort to move beyond the oversimplified M1/M2 framework. However, the manuscript still employs this terminology in a way that may appear contradictory to its main message. For example: “microglia in aged brains exhibit blunted responses to IL-4 stimulation, reflecting impaired polarization toward the M2 phenotype [22,23].”
Response 2: We appreciate this important point. To avoid relying on the M1/M2 framework, we revised the sentence to: “For instance, microglia in aged brains exhibit blunted responses to IL-4 stimulation, indicating an impaired ability to adopt anti-inflammatory or tissue-repair–associated states.” (p. 4, lines 128–130)
Comment 3: The subsection “2. Aging Microglia as Foes” is comparatively long and, in its current form, largely reiterates information that is already well established in the literature and covered in previous reviews. Considering that the core contribution of this article is the characterization of microglial subsets and state-specific changes. Authors should condense this section to avoid redundancy and to maintain focus on what is novel or particularly integrative in the present review.
Response 3: Thank you for pointing this out. We have substantially condensed Section 2 (“Aging Microglia as Foes”) to avoid redundancy and to remove content that is already well established in previous reviews. In particular, subsections 2.1–2.5 were reduced by approximately 40%. We revised the text by removing the step-by-step descriptions that listed previously reported microglial changes with multiple individual examples. Instead, we streamlined the section and replaced it with a concise summary that synthesizes the key phenomena in an integrated manner. We believe this restructuring improves clarity and maintains a stronger emphasis on the novel and integrative perspectives of this review.
Comment 4: The manuscript would be substantially strengthened by the inclusion of a summary table compiling key information for each disease or condition discussed.
Response 4: Thank you very much for this thoughtful suggestion. We fully agree that a comprehensive summary table would enhance clarity and accessibility for readers. In response, we have created a new summary table (p. 12, Table 2) that concisely compiles, for each disease or condition discussed, the key microglial subsets involved, their putative protective functions, potential detrimental changes, representative mechanisms, and hallmark references. This table allows readers to grasp the overall landscape at a glance, even without reading the full text, and we believe it substantially improves the readability and usefulness of the manuscript.
Comment 5: The use of dashes can add clarity and elegance to scientific writing when used to create emphasis or refine sentence rhythm. However, in the current version, dashes are employed excessively, which at times disrupts the flow and lends an overly informal tone to the text.
Response 5: Thank you very much for your stylistic suggestion. In response, we revised the manuscript and replaced unnecessary dashes with commas or periods where appropriate. This modification has improved readability while preserving the academic tone of the text (e.g., p.1, lines 10–12; p.1, line 33; p.2, line 44; p. 2, line 47-49; p. 2, line 62-64; p. 2, line 68-69; p. 2, line 88-90; p. 4, low 161-163; p. 4, low 169-176; p. 5, low 177; p. 5, low 200-202; p. 5, low 207-209; p. 8, low 256-259; p. 11, low 405-409; p. 11, low 415-419; p. 12, low 458-460; p. 14, low 513-516; p. 16, low 651-652; p. 17, low 677-682; p. 17, low 688-691; p. 19, low 740-743; p. 19, low 756-758; p. 21, low 835-838; p. 21, low 849-852).
Reviewer 2 Report
Comments and Suggestions for Authors
Microglia in Brain Aging and Age-Related Diseases: Friends or 2 Foes? This review summarizes evidence on microglial changes in aging and neurodegeneration, underlining their dualistic role and covers strategies that modulate microglial function to maintain brain health and prevent or treat frailty and age-related diseases. According to authors, the conclusions are that certain microglial populations adopt protective or adaptive phenotypes that preserve neural integrity while in circumstances of chronic inflammation or pathological conditions, even protective microglia may become inflammation-promoting. And so, the title is adequate to the research, discussion and conclusions drawn by the authors.
Detailed analysis:
Lines 33-37: "Microglia, the resident immune cells of the central nervous system (CNS), play indispensable roles in maintaining brain homeostasis through immune surveillance and the clearance of cellular debris. Under physiological conditions, microglia exhibit a highly dynamic and ramified morphology, constantly monitoring the neural environment to preserve tissue integrity." Please kindly add references for this sentence.
Lines 38-40: "However, with advancing age, microglia undergo profound morphological and functional alterations, including chronic activation, reduced motility, and dysregulated phagocytic activity." Please kindly add references for this sentence.
Lines 49-50: " Cognitive frailty represents a potentially reversible state of vulnerability that precedes irreversible neurodegeneration."Please kindly add references for this sentence.
Lines 60-70: "emphasizing protective subsets—such as CD11c⁺ microglia—that contribute to tissue repair and neural resilience". lease kindly add references for this sentence.
Lines 101 and 121: Please use italics for "in vivo".
Lines 204-205: " multicomponent programs can ameliorate or delay progression". Is it possible to give examples of types of multicomponent programs proposed by the authors? Please, include specific exercices and its references if possible.
Lines 245-246: "The traditional Kampo medicine Kai Xin-San has been shown to improve cognitive performance in SAMP8 mice". Please, provide the specific interventions used by Kampo medicine used to improve cognitive performance in SAMP8 mice and that activate the mentioned pathways.
Line 341: "Safaiyan et al." Please provide reference immediatly after the reference to the name. Eg.: "Safaiyan et al., [84] ..." The same for the reference "Krasemann et al." in line 391 - it should be mentioned as "Krasemann et al., [87]". Also, for " Benmamar-Badel et al." (line 494), "Jia et al." in line 683, " Kim et al. " (line 695), "Cao et al. " (line 699), Shen et al. (line731), "Berghoff et al. " (line 772), " Berghoff et al. " (line 808), "Nugent et al." (line 817), "Damisah et al." (line 828),
Line 889: Please delete A before Recently.
Author Response
We sincerely thank the reviewer for their thoughtful and constructive comments. We have carefully revised the manuscript in accordance with all suggestions. Below, we provide a point-by-point response. All changes have been highlighted in the revised manuscript.
Comment 1: Lines 33-38: "Microglia, the resident immune cells of the central nervous system (CNS), play indispensable roles in maintaining brain homeostasis through immune surveillance and the clearance of cellular debris. Under physiological conditions, microglia exhibit a highly dynamic and ramified morphology, constantly monitoring the neural environment to preserve tissue integrity." Please kindly add references for this sentence.
Response 1: Thank you for your suggestion. We added appropriate references supporting microglial surveillance, debris clearance, and homeostatic maintenance (p.1, low 33-38. Kettenmann et al., Physiological reviews, 2011; Prinz et al., Cell, 2019; Nimmerjahn et al., Science
, 2005).
Comment 2: Lines 38-40: "However, with advancing age, microglia undergo profound morphological and functional alterations, including chronic activation, reduced motility, and dysregulated phagocytic activity." Please kindly add references for this sentence.
Response 2: Thank you very much for this helpful comment. We have added multiple references describing age-associated morphological and functional changes (p.1, low 38-40.
Niraula et al., Neuropsychopharmacology, 2016; Mosher et al., Biochem Pharmacol, 2015; Streit et al., Glia, 2003).
Comment 3: Lines 49-50: " Cognitive frailty represents a potentially reversible state of vulnerability that precedes irreversible neurodegeneration. "Please kindly add references for this sentence.
Response 3: We appreciate your suggestion. We added references defining cognitive frailty and its reversibility (p.2, low 49-50. Kelaiditi et al., The Journal of nutrition, health & aging, 2013; Qingwei et al., Ageing Research Reviews, 2015).
Comment 4: lines 60-70: "emphasizing protective subsets—such as CD11c⁺ microglia—that contribute to tissue repair and neural resilience". lease kindly add references for this sentence.
Response 4: Thank you very much for this valuable suggestion. We have added several references supporting the involvement of CD11c⁺ microglia in repair, resilience, and lipid metabolism (p. 2, low 68-69. Sato-Hashimoto et al., eLife, 2019; Benmamar et al., Front. Immunol., 2020).
Comment 5: Lines 101 and 121: Please use italics for "in vivo".
Response 5: Thank you for pointing this out. We have corrected the formatting accordingly (p. 3, line 94, line 114). All occurrences of in vivo are now italicized throughout the manuscript.
Comment 6: Lines 204-205: " multicomponent programs can ameliorate or delay progression". Is it possible to give examples of types of multicomponent programs proposed by the authors? Please, include specific exercices and its references if possible.
Response 6: Thank you very much for this helpful suggestion. We have added examples of multicomponent interventions (aerobic exercise, strength and balance training, flexibility exercises, nutritional guidance, and cognitive training) and included relevant references (p. 4, low 163-168).
Comment 7: Lines 245-246: "The traditional Kampo medicine Kai Xin-San has been shown to improve cognitive performance in SAMP8 mice". Please, provide the specific interventions used by Kampo medicine used to improve cognitive performance in SAMP8 mice and that activate the mentioned pathways.
Response 7:
Thank you for this helpful comment. We have revised the text to clearly describe the specific intervention protocols used in SAMP8 mice, including the Kampo formula administered, its dosage, route of administration, and treatment duration. In addition, we added details on how the NLRP3/Caspase-1 pathway was identified—specifically, by incorporating inhibitor studies that revealed the pathway’s contribution to the cognitive effects. These revisions clarify the intervention methodology and explicitly highlight the mechanistic targets of the Kampo medicine. (p. 5, low 189-194)
Comment 8: Line 341: "Safaiyan et al." Please provide reference immediatly after the reference to the name. Eg.: "Safaiyan et al., [84] ..." The same for the reference "Krasemann et al." in line 391 - it should be mentioned as "Krasemann et al., [87]". Also, for " Benmamar-Badel et al." (line 494), "Jia et al." in line 683, " Kim et al. " (line 695), "Cao et al. " (line 699), Shen et al. (line731), "Berghoff et al. " (line 772), " Berghoff et al. " (line 808), "Nugent et al." (line 817), "Damisah et al." (line 828),
Response 8: Thank you for pointing this out. We have revised all instances the reviewer listed by adding the corresponding reference numbers immediately after each author name (e.g., “Safaiyan et al., [74]” low 246, low 714). This correction has been applied consistently throughout the manuscript, including the citations for Krasemann et al. (low 292), Benmamar-Badel et al. (low 395), Jia et al. (low 587), Kim et al. (low 599), Cao et al. (low 604), Shen et al. (low 636), Berghoff et al. (low 677), Nugent et al. (low 723), and Damisah et al. (low 734).
Comment 9: Line 889: Please delete A before Recently.
Response 9: Thank you for your careful reading. We have corrected the phrasing as requested by deleting the “A” before “Recently (p. 20, low 796).”